# Solar occultation measurement of mesospheric ozone by SAGE III/ISS: Impact of variations along the line of sight caused by photochemistry

**Murali Natarajan [1], Robert Damadeo [1], David Flittner [1]**

[1] Science Directorate, NASA Langley Research Center, 21 Langley Blvd., Mail Stop 401-B, Hampton, VA 23681, USA.

Correspondence to: Murali Natarajan (murali.natarajan@nasa.gov)

**Abstract.** Twilight gradients in the concentration of atmospheric species with short photochemical lifetimes influence the transmission data obtained in a solar occultation instrument like the Stratospheric Aerosol and Gas Experiment III aboard the International Space Station (SAGE III/ISS). These photochemically induced changes result in nonlinear asymmetries in the species distribution near the tangent altitude along the line of sight (LOS). The bias introduced by neglecting the effects of twilight variations in the retrieval of mesospheric ozone is the focus of this study. $O_3$ in the mesosphere exhibits large variations near the terminator during sunrise and sunset based on current understanding of the photochemistry of this altitude region. The algorithm used in the SAGE III/ISS standard retrieval procedure for mesospheric ozone does not include the effects of these gradients. This study illustrates a method for implementing a correction scheme to account for the twilight variations in mesospheric $O_3$ and gives an estimate of the bias in the standard retrieval. We use the results from a diurnal photochemical model

conducted at different altitudes to develop a database of ratios of mesospheric $O_3$ at different
solar zenith angles (SZA) around 90° to $O_3$ at a SZA of 90° for both sunrise and sunset
conditions. These ratios are used to scale the $O_3$ at levels above the tangent altitude for
appropriate SZA in the calculation of the optical depth along the LOS. In general, the impact of
the corrections due to twilight variations is to increase the contribution of the overlying layers to
the optical depth thereby reducing the retrieved $O_3$ concentration at the tangent altitude. We find
that at sunrise the retrieved mesospheric $O_3$ including the diurnal corrections is lower by more
than 30% compared to the archived $O_3$. We show the results obtained for different latitudes and
seasons. In addition, for nearly collocated sunrise and sunset scans, we note that these
corrections lead to better qualitative agreement in the sunrise to sunset $O_3$ ratio with the
photochemical model prediction.
**1  Introduction**
The solar occultation measurement technique has been the workhorse among various methods
used for monitoring the composition of the earth's atmosphere for over 4 decades. This is
evidenced by many successful experiments such as SAGE, SAGE II, Halogen Occultation
Experiment (HALOE), Atmospheric Trace Molecule Spectroscopy (ATMOS), Atmospheric
Chemistry Experiment – Fourier Transform Spectrometer (ACE-FTS), Polar Ozone and Aerosol
Measurement (POAM), SAGE III/M3M and SAGE III/ISS.  Major advantages of this technique
include high signal to noise ratio, high vertical resolution, and long-term accuracy provided by
the 'self-calibrating' nature of the instrument operation.  Limited global coverage ranks high
among the disadvantages of this method.  In the occultation experiments, the absorption of solar
radiance measured by the instrument as a function of tangent height altitude or pressure is related
to the optical depth and hence the abundance of the species along the line of sight (LOS). The
bulk of the absorption, in general, occurs around the tangent point because of the exponential
decrease in atmospheric density with altitude and due to the slant path determined by the
spherical geometry. Algorithms used in standard retrievals assume that the species distribution in
atmospheric layers is homogeneous and, therefore, the variation along the LOS is symmetrical
around the tangent point location.  The column along the LOS is then made up of species
concentrations at the tangent altitude and the layers above corresponding to a SZA of 90°.  This
assumption is quite valid for species such as $CH_4$, $H_2O$, and stratospheric $O_3$ because of their
long photochemical lifetimes and the absence of chemically induced diurnal variations. In the
case of species with short lifetimes, the sudden changes in the photolysis rates near day/night
terminator trigger rapid variations in the concentration as a function of SZA. These variations
result in nonlinear asymmetry along the LOS. In this case, the column along the LOS is made up
of species concentration at a SZA of 90° at the tangent altitude and those from the layers above
at SZA different from 90° on either side of the tangent point.

The influence of twilight variations in NO and ClO on the interpretation of solar occultation
measurements was described by Boughner et al. (1980).  Correction factors based on
photochemical models, as discussed in the above study, have been routinely applied in the
retrievals of stratospheric NO and $NO_2$ profiles in HALOE (Gordley et al., 1996; Russell et al.,
1988) and in ATMOS (Newchurch et al., 1996).  Brohede et al. (2007) described the role of
diurnal variations in the retrieval of $NO_2$ from OSIRIS measurements.  The algorithm used in the
retrieval of $NO_2$ in SAGE, SAGE II, SAGE III/M3M, and SAGE III/ISS neglects the twilight
variations.  A recent study of the $NO_2$ retrieval from SAGE III/ISS by Dubé et al. (2021)
describes the importance of considering the diurnal variations along the LOS.

Mesospheric $O_3$ is also characterized by short photochemical lifetimes and steep twilight
gradients and, therefore, it is a potential candidate species requiring appropriate corrections in a
retrieval from solar occultation instruments. Natarajan et al. (2005) noted that the diurnal
correction factors used in the retrieval of mesospheric ozone from HALOE (Version 19) needed
to be updated. They derived new factors from a diurnal photochemical model of mesospheric
ozone and illustrated the impact of the corrections using a small subset of retrieved HALOE
mesospheric $O_3$ profiles.  In the present study, we describe the application of similar corrections
to the SAGE III/ISS retrieval of mesospheric $O_3$. Table 1 of the Data Product User's Guide for
SAGE III/ISS (2021) lists the release status of mesospheric $O_3$ data as a Beta version that is yet
to be validated, because it is still potentially impacted by spectral stray light within the
instrument. Our goal is to quantify the impact of the corrections on the archived data and to see
whether the changes can support other known criteria. A description of the mesospheric $O_3$
variations under twilight conditions as calculated with a diurnal photochemical model is given in
section 2.  The occultation geometry and the diurnal correction factors for mesospheric $O_3$ are
described in section 3. Results from the application of the factors to correct the archived data are
discussed in section 4. We also include the results from an approximate retrieval using the
archived transmission data with and without diurnal corrections. A comparison of zonally
averaged $O_3$ profiles with scaled data for the same period from the Microwave Limb Sounder
(MLS) instrument on Aura satellite is described in the next section. This is followed by a
discussion of sunrise to sunset mesospheric $O_3$ ratios using appropriate collocated scans and a
comparison to theoretical values.  The final summary section reiterates the importance of
corrections for photochemically induced twilight mesospheric $O_3$ variations in solar occultation
retrievals.

**2   Mesospheric $O_3$ variations at sunrise/sunset**

We use a time-dependent, one-dimensional photochemical model to obtain the diurnal variation
in mesospheric $O_3$.  A detailed description of the model used in this study is given in Natarajan et
al. (2005).  This version of the model extends from 56 km to 100 km at 1 km intervals.  The
photochemical reaction scheme, shown in the Appendix, includes reactions involving species
from the Oxygen, Hydrogen, and Nitrogen families.  Chlorine and Bromine reactions do not play
a significant role in this region of the atmosphere. The adopted chemical rate constant data are
from the JPL Publication 19-5 (2020). The diurnal model does not use a family approximation
and reactive species O, $O_3$, N, NO, $NO_2$, H, OH, $HO_2$, and $H_2O_2$ are considered as independent
variables. The concentrations of long-lived species are constrained by the results from a two-
dimensional chemical transport model (CTM) (Callis et al., 1997).  Diffusion coefficients from
the CTM are used to parameterize the vertical transport.  The diurnal model uses a variable time
step, variable order stiff equation solver (Byrne and Hindmarsh, 1975) to integrate the system of
species continuity equations. The maximum time step is 600 seconds, and the algorithm
automatically reduces the time step to very low values if needed. The model is run for 4 diurnal
cycles so that the reactive species reach a steady diurnal behavior, and the results from the fifth
cycle are used in the analysis.  The model is run for every month at 11 latitudes, corresponding to
the latitude nodes of the CTM, from 56.25° N to 56.25° S at an interval of 11.25°.

Calculated $O_3$ diurnal variation in June at the latitude of 11.25° S and at different altitudes of

interest to this work is illustrated in Fig.1. We are restricting our attention to altitudes below 74

km because the SAGE III/ISS $O_3$ data are noisy in the region above and the quoted uncertainty is

also large. $O_3$ concentration is shown as a function of time starting at midnight. Nighttime $O_3$

has a constant value representing the total odd oxygen in the lower mesosphere. A sharp

decrease at sunrise is mainly caused by photolysis of $O_3$ forming atomic oxygen. The

recombination of atomic oxygen and $O_2$ quickly balances the loss of $O_3$ from photolysis. This

reaction is pressure dependent and becomes slower at higher altitudes. The photolysis of $O_2$

generates additional odd oxygen ($O_X = O + O_3$) and in the morning hours this leads to an increase

in both $O_X$ and $O_3$. The formation of odd hydrogen species from the reaction of $O(^1D)$ with $H_2O$

during the day triggers the catalytic destruction of odd oxygen through reactions involving OH.

It is noted that between 50 and 80 km the chemical time constant of $O_x$ is of the order of few

hours and $O_x$ exhibits a diurnal variation caused by the competing production and destruction

reactions. In the early morning there is a net gain of $O_X$ and in the evening, there is net loss of

$O_X$, which continues even after sunset until atomic oxygen is depleted. The partitioning of Ox

into O and $O_3$ is mainly controlled by the photolysis of $O_3$ and the production of $O_3$ through the

recombination of O and $O_2$. The large increase in $O_3$ seen around sunset is mainly due to the

decrease in the photolysis of $O_3$ and the continuation of the recombination of O and $O_2$. $O_3$

reaches a steady value within an hour or so after sunset. The diurnal model extends to 100 km;

however, since the quoted uncertainty above 70 km in the archived SAGE III/ISS $O_3$ is large, we

will focus on the region below.


The results of the full diurnal cycle are of general interest about the model simulation. But, with
reference to solar occultation measurements, the sharp gradients seen in the $O_3$ concentration
near SZA of 90° are more critical.  The significance of the twilight variations to the retrieval of
mesospheric $O_3$ under sunrise/sunset conditions can be understood with the help of the schematic
shown in Fig.2. This illustrates the occultation geometry in the plane containing the LOS.  The
red line denotes the LOS at a tangent altitude of $Z_T$.  Points F and N represent the intersection of
the LOS with an atmospheric layer at an altitude of Z shown in green.  For a species with little or
no twilight variations, the concentrations at the locations F and N are nearly equal to that at the
location U, the tangent point at an altitude of Z.  In this case, the concentrations at tangent height
$Z_T$ can be derived in a straightforward manner from the measured transmission using a retrieval
algorithm. However, if the photochemistry causes significant gradients near SZA of 90°, as in
the case of mesospheric $O_3$, the distribution around the tangent point becomes nonlinearly
asymmetric because the concentrations at F and N depend on the respective local SZA. This
variation must be incorporated in the evaluation of $O_3$ specific optical depth along the LOS.

To illustrate the impact of diurnal variations on slant-path column of $O_3$, we selected a typical
event from the SAGE III/ISS data and applied the calculated $O_3$ variations in the slant-path
column evaluation.  The required parameters include month, date, event type (sunrise or sunset),
tangent altitude, latitude, longitude, spacecraft latitude and longitude. These data are taken from
the current Version 5.2 SAGE III/ISS data available from the Atmospheric Sciences Data Center
(ASDC) at NASA Langley Research Center. We used the model results for June at 11.25° S
latitude to get the $O_3$ variation at sunrise along the LOS corresponding to different tangent
altitudes from 56 to 76 km. The latitude of the chosen SAGE III/ISS measurement is 11.35° S.
The $O_3$ concentration along the LOS for tangent altitude of 64 km is shown as a function of
distance along the LOS relative to the tangent point in the left panel of Fig.3. The dotted line
corresponds to the $O_3$ concentration along the LOS when the diurnal variations are neglected and
only the values corresponding to 90° SZA from the layers above the tangent altitude are used.
The solid line represents the $O_3$ including the diurnal variations at the respective altitudes. The
increased $O_3$ concentrations on the instrument side of the LOS are readily seen. The ratio of the
$O_3$ column along the LOS with diurnal variations to the column without the diurnal variations is
shown as a function of tangent altitude in the panel on the right side. The peak difference of the
order of 30% occurs in the altitude range from 61 to 72 km. Underestimation of the partial $O_3$
slant-path column from layers above the tangent altitude in the standard retrieval translates to
overestimation of the retrieved $O_3$ at the tangent altitude. The bias introduced by the neglect of
twilight variations can be evaluated with the help of the diurnal model results.

The technique is to express the $O_3$ variation as a function of SZA in terms of concentration
normalized to $O_3$ at SZA of 90°.  Figure 4 shows the distribution of the ratio $O_3(\theta)/O_3(\theta=90°)$
near sunrise as a function of SZA and altitude obtained from the model results for 11.25° S
latitude in June. For a given tangent height, the total slant-path $O_3$ column comprises of partial
slant-path columns corresponding to the layers at and above the tangent height. Spherical
geometry dictates that the partial pathlength along the LOS is maximum for the layer
immediately above the tangent height (i.e., the lowest layer) and decreases dramatically for
higher layers. This, combined with decreasing $O_3$ concentration with height in the lower
mesosphere, results in a total slant-path column dominated by contributions from a few layers
right above the tangent point.  Therefore, only a small range of SZA, say between 86° and 94°,
centered at 90° are important.  At 62 km the $O_3$ ratio is less than 1.0 for SZA less than 90° and it
increases gradually for SZA greater than 90°.  At higher altitudes, the ratio shows a much steeper
increase for SZA greater than 92°. The ratio, in some cases, is even slightly larger than 1.0 at
SZA less than 90°.  From the occultation geometry shown in Fig.2, it is seen that as one moves
away from a SZA of 90° along the LOS at any tangent altitude, the corresponding altitude layer
of interest moves upwards.  Figure 5 illustrates the $O_3$ twilight ratio as a function of SZA and
altitude for sunset conditions for the same latitude and month. The changes in the ratio for sunset
condition are smaller and more gradual especially for SZA greater than 90° compared to the
sunrise case.  It should be recalled that the daytime variation in the odd oxygen concentration in
the lower mesosphere impacts the $O_3$ concentration differently at sunrise and sunset. The
differences between the $O_3$ variations for sunrise and sunset conditions suggest that the effects on
the retrievals are different for sunrise and sunset occultations. The twilight $O_3$ ratios for altitude
layers above the tangent altitude can be used to get the $O_3$ concentration and hence the optical
depth along the LOS more accurately.

Mesospheric $O_3$ concentrations are influenced by reactions involving $HO_x$ species and therefore
the distribution of $H_2O$ used in the model is an important factor.  An earlier study with HALOE
mesospheric $O_3$ data (Natarajan et al. 2005) using the results from the same CTM showed that
the monthly, zonal mean $H_2O$ distribution from the CTM was in good agreement with the data
taken from the UARS reference atmosphere project.  Linear trend in mesospheric $H_2O$ and solar
cycle response have been addressed in literature (Remsberg et al., 2018; Yue et al., 2019).  Yue
et al. (2019) report a trend in mesospheric $H_2O$ of the order of 4 to 6% per decade based on the
data from the Sounding of the Atmosphere using Broadband Emission Radiometry (SABER) and
MLS instruments.  Long term variability in $H_2O$ certainly impacts the absolute level of
mesospheric $O_3$.  But, for the present study, the factor of importance is the relative variation of
$O_3$ very close to SZA of 90° during sunrise and sunset in the mesosphere.  We have done a
sensitivity study at 11.25° S in June using the diurnal model with a 25% increase in the $H_2O$
concentration.  Figure 6 displays the percent change in the twilight $O_3$ ratios for sunrise shown in
Fig.4.   The maximum impact below 74 km is less than 20% and it is very small in the lower
regions.  The twilight ratio in $O_3$ is quite robust and small changes in the atmospheric parameters
such as temperature and $H_2O$ do not impact this ratio much. The use of this ratio is a valid
approximation in correcting the retrieval scheme.

**3 SAGE III/ISS Mesospheric ozone**

The Sage III/ISS instrument payload was launched in February 2017 and successfully attached to
the ISS. The ISS occupies a low earth orbit at an inclination of 51.64° that provides occultation
coverage of low- and mid-latitude regions.  Description of the experiment and early validation of
the $O_3$ measurements are given in McCormick et al. (2020) and Wang et al. (2020). More
detailed information on the various wavelength channels and data used for retrieving a suite of
atmospheric species including mesospheric $O_3$ are given in SAGE III Algorithm Theoretical
Basis Document (ATBD, 2002) and in the SAGE III/ISS Data Products User's Guide Version
3.0 (2021) (DPUG). Among the three different $O_3$ profile measurements made by the instrument,
the one based on short wavelengths in the Hartley-Huggins bands refers exclusively to
mesospheric $O_3$. Three Charge-Coupled Device (CCD) pixel groups (PGs 0-2) are assigned to
the short wavelengths in the 280-293 nm range, though only one (PG 1 centered at 286 nm) is
currently used for the retrieval.  According to the DPUG, mesospheric $O_3$ data have not been
fully validated. We also note that the uncertainty in the archived $O_3$ concentration becomes
larger than 10% above 70 km and there are some spurious negative data pointing to uncertainties
in the transmission. The present study focusses only on SAGE III/ISS $O_3$ in the lower
mesosphere up to an altitude of 70 km even though the retrieval itself starts at 90 km.  The
diurnal model described in the previous section extends up to 100 km. We use the Version 5.2
transmission and species data obtained from ASDC at NASA LaRC. For each year and month,
we have categorized the scans according to event type, sunrise, or sunset. The input data for our
analysis include the tangent point latitude and longitude, spacecraft latitude and longitude,
vertical profiles of neutral density, mesospheric $O_3$, and transmission. We use only the
transmission data from the science pixel group 1 (PG1), which has a center wavelength of
286.124 nm, since the predominant species active in this wavelength region is $O_3$.

We have generated a database of $O_3$ twilight ratios for sunrise and sunset conditions from the
diurnal model results. These ratios cover for each month the latitude range from 56.25° N to
56.25° S at an interval of 11.25°, SZA from 84° to 96° at 0.5° intervals, and altitudes from 56 to
90 km at 0.5 km intervals. Using the input data from each of the SAGE III/ISS occultations and
spherical geometry relations, for every tangent altitude we compute the SZA as well as partial
pathlengths corresponding to overlying layers. This generates a pathlength matrix like the one
used in the standard retrieval.  Appropriate $O_3$ twilight ratios are then obtained by interpolation
using the SZA and layer altitude. Multiplication of the standard pathlength matrix by the $O_3$
ratios yields the modified pathlength matrix including the effects of diurnal variations.

The twilight ratios can either be used to modify the $O_3$ profiles from the standard retrieval or be
incorporated in a new retrieval from measured transmission profile. The first method is like the
procedure described by Dubé et al. (2021) for making diurnal corrections to stratospheric $NO_2$
data from SAGE III/ISS. The archived SAGE III/ISS $O_3$ profile and the standard pathlength
matrix are used to recreate the $O_3$ specific slant optical depth, as shown by the equation
$$\tau = \sigma\, S\, n, \qquad\qquad (1)$$
where $\tau$ is the $O_3$ slant optical depth profile, $\sigma$ is the $O_3$ cross section corresponding to the center
wavelength of PG1, and n is the $O_3$ profile from the standard retrieval. S represents the
pathlength matrix with each row corresponding to a tangent point altitude. This can be written as
a triangular matrix because of the geometric symmetry on opposite sides of the tangent point as
can be seen from Fig.2. The slant optical depth can then be converted to a $O_3$ vertical profile
corrected for diurnal variations using the modified pathlength matrix described earlier, as shown
by the equation
$$n_{wd} = (S_{wd})^{-1}\, \tau/\sigma = (S_{wd})^{-1}\, S\, n \qquad\qquad (2)$$
where $S_{wd}$ is the modified pathlength matrix with diurnal correction and $n_{wd}$ is the corrected $O_3$
profile. Here it is assumed that the $O_3$ absorption coefficient remains constant along the LOS.
This procedure gives a quantitative estimate of the over-prediction by the standard retrieval. The
results for a sunrise event on June 14, 2021 (Event ID =2021061438SR) are shown in Fig.7. The
left panel displays the $O_3$ concentration profiles – the solid red line is the archived data from
standard retrieval and the solid black line represents the profile after applying the diurnal
correction ratios to the pathlength matrix. The percent difference between the standard and the
modified profiles is shown by the solid line on the right panel. For this occultation, the difference
exceeds 40% above 64 km. This is consistent with the change in $O_3$ slant column due to the
diurnal correction shown in Fig.3. We also note that the retrieval becomes noisy in the upper
altitudes as $O_3$ concentrations reach near detection limits. In the second method, instead of
evaluating the slant optical depth using Eq. (1), the archived slant-path transmission data, which
corresponds to PG1, is used along with the standard and modified pathlength matrices to retrieve
the vertical $O_3$ profiles. The change in the slant-path transmission corresponding to the science
CCD channel PG1 for each tangent altitude below an upper boundary of 90 km is related to the
total slant optical depth made up mainly of $O_3$ absorption and Rayleigh scattering contributions.
After removing the Rayleigh scattering part corresponding to the center wavelength of 286.124
nm, the slant-path $O_3$ column can be estimated using the $O_3$ absorption coefficient at this
wavelength taken from Bogumil et al. (2003), which is the same database used in the SAGE
retrieval algorithm. The standard and modified pathlength matrices are then used to get the
vertical $O_3$ profiles without and with corrections for diurnal variations respectively. The retrieved
$O_3$ profiles for the sunrise event mentioned earlier are given by the dashed lines on the left panel
of Fig.7, the red color denoting the standard retrieval without diurnal corrections and the black
color the modified retrieval with diurnal corrections. We have used a very simple algorithm and
assumed that the transmission data corresponds to a single wavelength to simplify the
calculation. The actual retrieval procedure used for the archived products may have included
more refinements. The agreement between results of the two different methods is very good,
both for the vertical $O_3$ profiles and for the percent differences. Results for a sunset event, closer
to the above sunrise event in location and within a day (Sunset event ID = 2021061515SS) are
shown in Fig.8. The impact of the diurnal correction is much smaller for sunset conditions. The
maximum difference between the standard and modified profiles is less than 10%. The two
different procedures for incorporating diurnal effects yield very nearly same results.

We have applied the diurnal corrections following the procedure described above to all the
SAGE III/ISS measurements from June 2021, categorized by the event type of sunrise or sunset.
Individual $O_3$ profiles were grouped together in 11 latitude bands, 11.25° wide between 56.25° N
and 56.25° S. The percent difference between the standard retrieval profile and the
corresponding modified profile, defined as $(O_3/O_{3,\ WD} - 1)$ *100, was calculated and the mean for
each latitude band was evaluated. The subscript WD refers to the retrieval including the diurnal
corrections. Figure 9 shows the resulting distribution of the mean, which represents the over-
estimation by the standard retrieval, as a function of latitude and altitude. There is a latitudinal
dependence with peak values occurring near 64 km and the summer hemisphere showing smaller
difference. Values higher than 100% (dark violet region) are seen in the upper altitudes of the
winter hemisphere. The $O_3$ profile has a sharp gradient reaching a very low minimum in winter
between and 70 and 80 km.  The retrieved data in this region are very noisy and thus sometimes
include negative concentrations.  The percent difference between the two retrievals also displays
a very noisy distribution with large values of both signs.  At altitudes above 85 km, the day-night
terminator occurs at solar SZA greater than 96° and $O_3$ variation around 90° is small. The bias in
the standard retrieval (not shown) is also small and there is no need for diurnal correction. The
distribution of percent differences for sunset measurements is shown in Fig.10. The values are
much smaller as discussed earlier, since the diurnal corrections are not significant for sunset. To
look at the seasonal dependence of the impact of diurnal corrections on the retrieved $O_3$, we have
repeated the procedure with SAGE III/ISS data from January 2021. Figure 11 displays the results
for sunrise conditions.  The differences between the standard and modified retrievals are larger
again in the winter hemisphere with peak values occurring near 64 km.  This agrees qualitatively
with Fig.11 of Natarajan et al. (2005), which showed the percent difference in retrieved HALOE
sunrise $O_3$ for January 1995 with updated diurnal correction factors compared to the retrieval
with HALOE version 19 correction factors. The archived HALOE version 19 retrieval used
correction factors from a diurnal calculation at 61 km for all mesospheric tangent altitudes
above. Since a partially corrected (version 19) retrieval was used as the basis, the contour levels
are negative and smaller in magnitude.

**4 Comparisons with other measurements**

It is of interest to see whether the correction to the retrieval of mesospheric ozone described
above can be validated by comparisons with other independent measurements. Mesospheric
ozone mixing ratios at SZA of 90° during sunrise and sunset have been measured by other solar
occultation experiments like HALOE and ACE-FTS. HALOE version 19 retrievals use
correction factors based on diurnal model calculation near the stratopause.  An update to these
correction factors was discussed in Natarajan et al. (2005) but a modified version of the full
ozone dataset was not generated. As far as we know, the retrieval scheme of ACE-FTS does not
use any correction for twilight variations of mesospheric ozone. It should be emphasized that
comparisons with data from other solar occultation experiments do not necessarily provide a
robust independent validation of the need to make such corrections to reduce the bias in the
measurements.

MLS aboard the Aura satellite also provides vertical profiles of $O_3$ extending into the
mesosphere. MLS measurements occur twice a day, once in the early afternoon and the other
past midnight.  Strode et al. (2022) have used the MLS data scaled with factors derived from
Goddard Earth Observing System (GEOS) model coupled with the Global Modeling Initiative
(GMI) chemistry mechanism for comparisons with SAGE III/ISS $O_3$ in the stratosphere. We
have done similar comparisons for a selected subset of the data in the lower mesosphere using
the results from the mesospheric diurnal model described earlier. We limited our attention to the
data in altitude range from 56 to 70 km.  We used the information provided in the MLS-V5 data
quality document (Livesey et al., 2022) to properly screen the $O_3$ data. The vertical resolution for
MLS $O_3$ varies from 3 to 5.5 km in the lower mesosphere. The reported accuracy varies from 8%
at 0.21 hPa to 40% at 0.02 hPa. We used the MLS V-5 $O_3$ profiles from a 11.25° latitude band
centered at 11.25° S from June 13 to June 15 of 2021. The native units of MLS measurements
are mixing ratios on pressure levels. We used the MLS temperature and geopotential height data
to get $O_3$ concentrations on an altitude grid. We derived the mean and the standard deviation
profiles for both day and night MLS measurements. Results from diurnal model calculations
were used to convert MLS day and night measurements to SZA of 90 ° during sunrise and sunset
conditions. Figure 12 shows the $O_3$ concentration at sunrise based on MLS night data by
asterisks and that based on MLS day data by diamonds. The horizontal lines represent the
standard deviations at different altitudes. We also obtained the mean and standard deviation
profiles using SAGE III/ISS data from the same latitude band and period in June 2021 like the
selected MLS data. The solid black line in Fig. 12 shows the mean sunrise profile from standard
retrieval and the standard deviation is represented by the yellow color band.  The dashed black
line is the modified retrieval with the green band showing the standard deviation.  The twilight
corrections to the mesospheric $O_3$ retrieval brings the profile in better agreement with that
derived from MLS day and night data. Above 68 km the MLS day measurements have large
variability, and the standard deviation is larger than the mean. Figure 13 shows the comparison
of the profiles for sunset conditions. The difference between the modified and the standard
retrievals is much smaller for the sunset conditions compared to the sunrise condition. Overall
SAGEIII/ISS mesospheric $O_3$ has a positive bias. The vertical resolution of SAGE III/ISS data is
about 0.7 km which is finer than the MLS vertical resolution.  We found that the application of
the MLS $O_3$ averaging kernel to smooth the SAGE III/ISS data has a minimum impact on the
comparison.

There have been several ground-based microwave measurements of atmospheric $O_3$ and its
diurnal variations (Connor et al., 1994; Parrish et al., 2014; Sauvageat et al., 2022). The
microwave radiometry (MWR) in Switzerland (Sauvageat et al., 2022) provides data temporally
overlapping the SAGE III/ISS data. These data are from measurements made at 2 ground stations
and they extend into the mesosphere. The vertical resolution of ground based MWR is very
coarse in the lower mesosphere, about 17 km (Connor et al., 1994). Therefore SAGE III/ISS $O_3$
data should be convolved with the averaging kernels of MWR prior to comparisons. In addition,
MWR provides hourly data and, unless the local measurement time coincides with SZA of 90°
during sunrise and sunset, the data must be converted using factors based on diurnal model. We
feel that comparison with MWR data is outside the scope of this paper.

**5 Sunrise to Sunset Ratio**

Brühl et al. (1996), in their paper on HALOE $O_3$ channel validation, discussed the sunrise to

sunset differences in $O_3$ around 0.1 hPa (about 64 km). Mesospheric layers are under sunlit

conditions even at SZA slightly greater than 90° at dawn and dusk.  As explained earlier, the

viewing geometry in solar occultation observations leads to an increase in the contribution of

overlying layers to the $O_3$ optical depth because $O_3$ concentrations corresponding to varying SZA

greater than 90° are seen along the LOS.  We have noted that the impact is larger during sunrise

than sunset measurement.  The sunrise to sunset $O_3$ concentration ratio becomes larger if the

diurnal variations along the LOS are not considered in the retrieval. Solar occultation

experiments occasionally offer the opportunity to approximately check this ratio as a test of

consistency of measurement and agreement with theory.  This is possible when sunrise and

sunset orbits cross over each other within a reasonably short interval of time and physical

proximity.  Such near coincidences are quite rare. We selected sunrise/sunset pairs of

measurements by SAGE III/ISS having tangent locations within 1.5° latitude, and 15° longitude

of each other and separated by a maximum of 36 hours. The effect of advection by the prevailing

westerly wind requires that the time and longitude differences are in the correct direction. There

are just 10 pairs of sunrise /sunset measurements in June 2021 that satisfy the above criteria, all

of them in low latitudes with a mean latitude of 10.46° S at sunrise and 10.27° S at sunset. The

mean of the sunrise to sunset ratios of $O_3$ concentrations from these scans is shown in Fig.14.

The solid line corresponding to the standard retrieval shows ratios greater than 1.1 above 60 km.

The green color shade represents the standard deviation. The modified retrieval yields a ratio

shown by the dashed line decreasing from 1.01 at 60 km to lower values above.  The horizontal

lines are the standard deviations.  The asterisk symbols represent the ratio from the diurnal

model.  The model value is in good agreement with the ratios from both the standard and
modified retrievals near 58 km but above this altitude there is some difference. The variation
with altitude in the model ratio is more like that shown by the modified retrieval.  The modified
retrieval qualitatively reflects the pattern that photochemistry of $O_3$ suggests in this altitude
region.  This comparison serves as an independent criterion to highlight the importance of
including the LOS twilight variations in the retrieval of mesospheric $O_3$ in solar occultation
measurements.  We noticed that very few such pairs of measurements, which satisfied the criteria
we have chosen, occurred during other months in SAGE III/ISS data. We have also looked at the
latitudinally averaged sunrise and sunset data for June 2021 obtained for generating figures 9 and
10.  For the latitude band centered at 11.25° S, the sunrise to sunset ratio as a function altitude
(not shown) is like Fig.14, which used only collocated data.  The small sampling size of the
collocated pairs of data and regions of overlapping standard deviations seen in the Fig.14 make
this at best an approximate comparison.  Other independent measurements are needed to verify
the altitude variation of the ratio of sunrise to sunset $O_3$ concentrations.

**6 Summary**

Photochemically induced changes in species concentration at twilight can cause asymmetries in
the distribution along the LOS of a solar occultation observation, variations that must be
considered in the retrieval algorithm. Prominent among the species that need corrections for
twilight variations are NO and $NO_2$ in the stratosphere and $O_3$ in the mesosphere.  The SAGE
III/ISS instrument uses the measurements in the short-wave Hartley-Huggins band to get
mesospheric $O_3$ profiles. The standard retrieval procedure does not consider the LOS variations
in $O_3$ caused by photochemistry. This study describes a procedure to use results from diurnal
photochemical model simulations to develop correction factors for different altitudes, latitudes,
and months. These factors were used along with the archived SAGE III/ISS mesospheric $O_3$ data
for selected time periods to obtain modified $O_3$ profiles. For the month of June 2021, it is shown
that neglecting the diurnal variations can result in nearly 50% overestimation of $O_3$ at 64 km at
lower latitudes. An approximate retrieval using the transmission data from SAGE III/ISS also
indicates similar behavior in the profiles obtained with and without diurnal corrections. The
retrievals were repeated for January 2021 to study the seasonal impact. Larger differences are
generally seen near 70 km in high latitude winter hemisphere, and this is most likely due to a
combination of very low $O_3$ concentrations, large twilight correction factors, and large
uncertainties in the data. The results from this study are in good agreement with those obtained
for the retrieval of HALOE mesospheric $O_3$ data.

SAGE III/ISS data include a few nearly collocated sunrise and sunset measurements, mostly in
the low latitudes and about a day apart. There are 10 pairs of such sunrise and sunset
measurements in June 2021. An analysis of the sunrise to sunset ratio profile from these data
indicates that the retrievals that include the diurnal variations show qualitatively better agreement
with theoretical prediction.

**Appendix**

454       Photochemical reactions considered in the mesospheric diurnal model:


(1)    $O_2$    +    $h\upsilon$    →    O    +    O
(2)    $O_3$    +    $h\upsilon$    →    O    +    $O_2$
(3)    $O_3$    +    $h\upsilon$    →    $O(^1D)$    +    $O_2$
(4)    $NO_2$    +    $h\upsilon$    →    O    +    NO
(5)    $H_2O$    +    $h\upsilon$    →    OH    +    H
(6)    $H_2O_2$    +    $h\upsilon$    →    OH    +    OH
(7)    NO    +    $h\upsilon$    →    N    +    O
(8)    $H_2O$    +    $h\upsilon$    →    $H_2$    +    O
(9)    $O(^1D)$    +    $O_2$    →    O    +    $O_2$
(10)    $O(^1D)$    +    $N_2$    →    O    +    $N_2$
(11)    $O(^1D)$    +    $H_2O$    →    OH    +    OH
(12)    O    +    O    +    M    →    $O_2$    +    M
(13)    O    +    $O_2$    +    M    →    $O_3$    +    M
(14)    O    +    $O_3$    →    $O_2$    +    $O_2$
(15)    O    +    OH    →    H    +    $O_2$
(16)    O    +    $HO_2$    →    OH    +    $O_2$
(17)    $NO_2$    +    O    →    NO    +    $O_2$

| 473 | (18) | H | + | $O_2$ | + | M | $\rightarrow$ | $HO_2$ | + | M |
|---|---|---|---|---|---|---|---|---|---|---|
| 474 | (19) | $O_3$ | + | OH | | | $\rightarrow$ | $HO_2$ | + | $O_2$ |
| 475 | (20) | $O_3$ | + | NO | | | $\rightarrow$ | $NO_2$ | + | $O_2$ |
| 476 | (21) | $O_3$ | + | H | | | $\rightarrow$ | OH | + | $O_2$ |
| 477 | (22) | OH | + | $HO_2$ | | | $\rightarrow$ | $H_2O$ | + | $O_2$ |
| 478 | (23) | $HO_2$ | + | NO | | | $\rightarrow$ | OH | + | $NO_2$ |
| 479 | (24) | OH | + | $H_2O_2$ | | | $\rightarrow$ | $H_2O$ | + | $HO_2$ |
| 480 | (25) | $HO_2$ | + | $HO_2$ | | | $\rightarrow$ | $H_2O_2$ | + | $O_2$ |
| 481 | (26) | $HO_2$ | + | $O_3$ | | | $\rightarrow$ | OH | + | $2 O_2$ |
| 482 | (27) | $O(^1D)$ | + | $H_2$ | | | $\rightarrow$ | OH | + | H |
| 483 | (28) | N | + | $O_2$ | | | $\rightarrow$ | NO | + | O |
| 484 | (29) | N | + | NO | | | $\rightarrow$ | $N_2$ | + | O |
| 485 | (30) | N | + | $NO_2$ | | | $\rightarrow$ | $N_2O$ | + | O |
| 486 | (31) | H | + | $HO_2$ | | | $\rightarrow$ | OH | + | OH |
| 487 | (32) | H | + | $HO_2$ | | | $\rightarrow$ | $H_2O$ | + | O |
| 488 | (33) | H | + | $HO_2$ | | | $\rightarrow$ | $H_2$ | + | $O_2$ |
| 489 | (34) | OH | + | $H_2$ | | | $\rightarrow$ | $H_2O$ | + | H |

**Data Availability**
SAGE III/ISS version 5.2 data is available from https://asdc.larc.nasa.gov/project/SAGE%20III-
ISS/g3bssp_52. MLS $O_3$ data are available from https://disc.gsfc.nasa.gov/. $O_3$ twilight ratios
used in this study are available from the author. They can also be obtained from any diurnal
photochemical model of the mesosphere.

**Author Contribution**
MN conducted the photochemical model calculations, SAGE III/ISS $O_3$ retrievals, and the
analyses described in the study, and he wrote the manuscript. RD and DF provided information
and guidance on the use of SAGE III/ISS mesospheric $O_3$ data as well as comments on the
manuscript.

**Competing Interests**
The authors declare that there is no competing interest for this study.

**Acknowledgements**
SAGE III/ISS data used in this study were obtained from the NASA Langley Research Center
Atmospheric Science Data Center. MN carried out this work while serving as a Distinguished
Research Associate of the Science Directorate at NASA Langley Research Center. MN thanks
Ellis Remsberg for reading and commenting on the draft version of this manuscript.

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

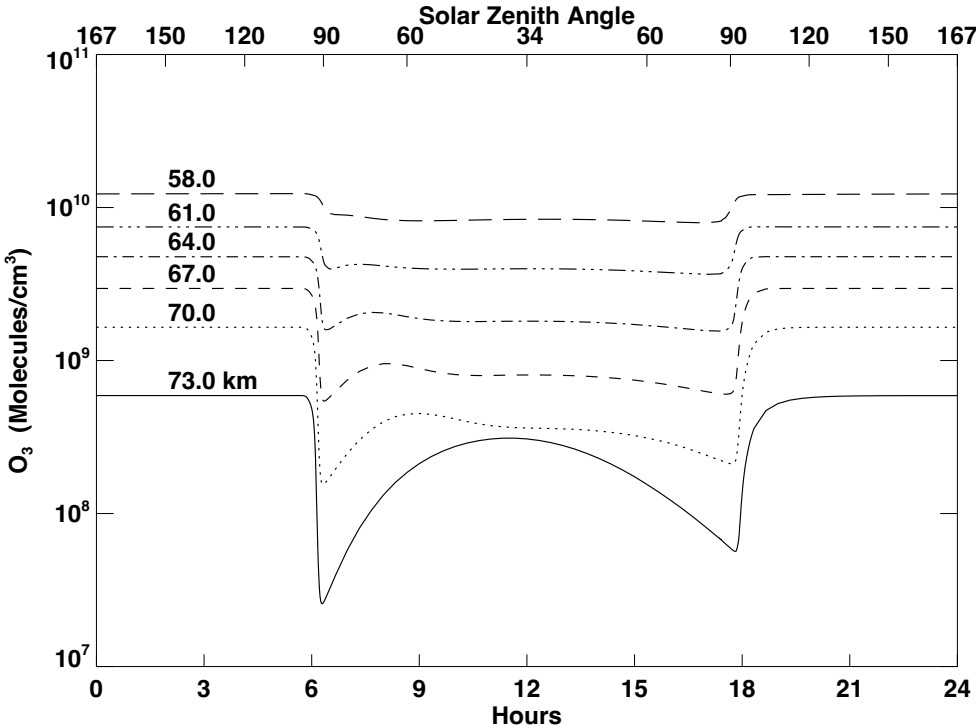


Figure 1.  Diurnal variation in $O_3$ at 11.25° S in June at altitudes from 58 to 73 km. 0 hours
denote midnight. The upper X axis shows the variation of SZA.

**Solar Occultation Geometry**

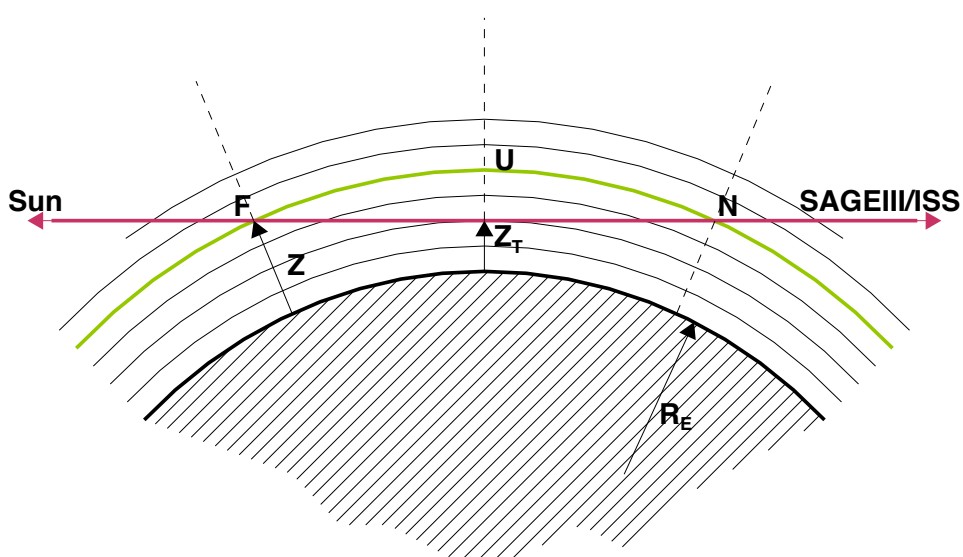

$Z_T$ **Tangent altitude, Z Altitude of a layer above the tangent path**

$R_E$ **Earth radius**


Figure 2.  Schematic representation of the solar occultation measurement. $Z_T$ is the tangent
altitude, red line is the LOS, Z is the altitude of a layer above the tangent altitude, F (towards
sun) and N (towards SAGE III/ISS) are the points of intersection of layer at Z with the LOS, and
$R_E$ is the earth radius.

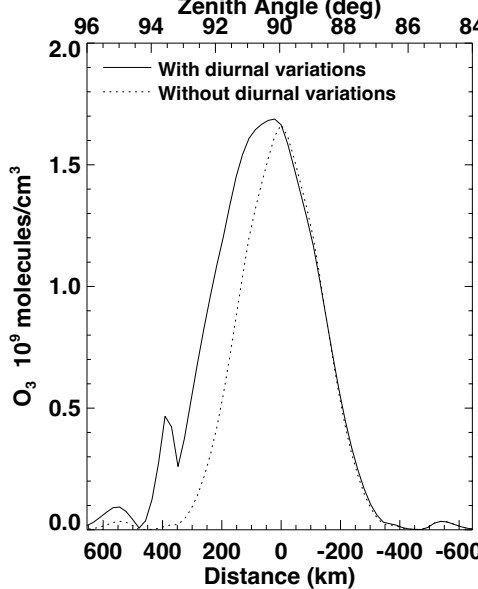

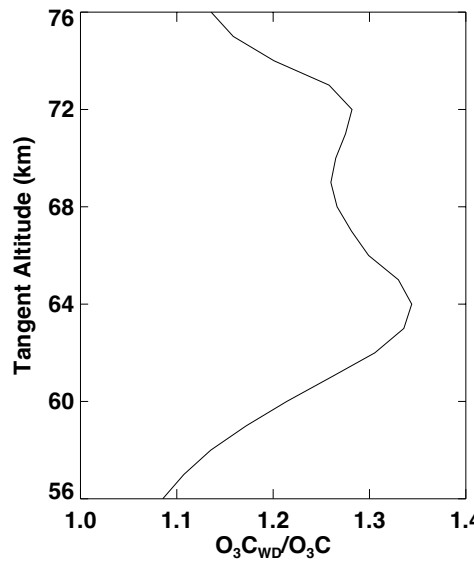


Figure 3. (Left) $O_3$ concentration along the LOS for a tangent altitude of 64 km at sunrise at
11.25° S latitude in June. Solid line shows $O_3$ with diurnal variations and the dotted line
represents $O_3$ without diurnal variations. The X-axis represents the distance along the LOS
relative to the tangent point with positive direction towards the instrument and negative direction
towards the Sun.  The upper axis shows the corresponding SZA. (Right) Ratio of the $O_3$ column
along the LOS with appropriate diurnal variations to the $O_3$ column without diurnal variations,
plotted as a function of altitude at 11.25° S in June.



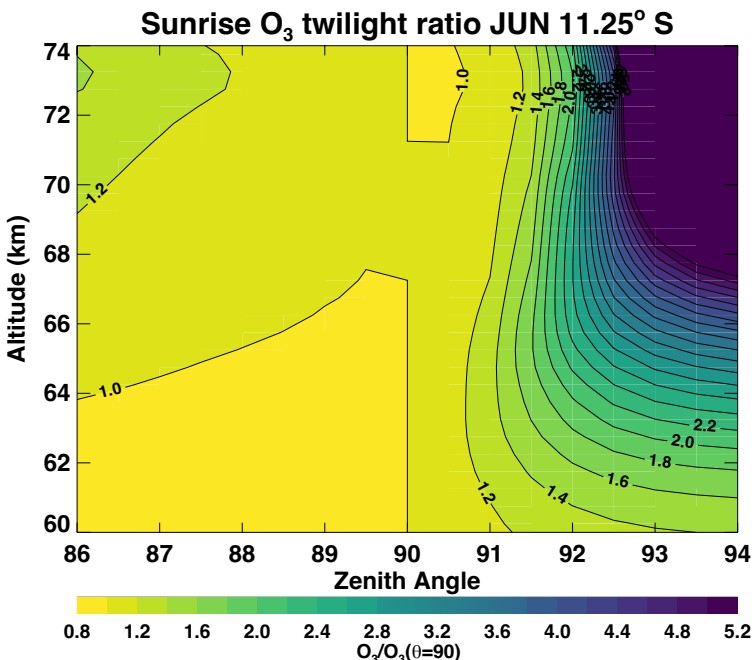


Figure 4.  Ozone twilight ratio, defined as $O_3$ at solar zenith angle $\theta/O_3$ at $\theta=90°$, as a function
of SZA and altitude for sunrise in June and 11.25° S latitude.


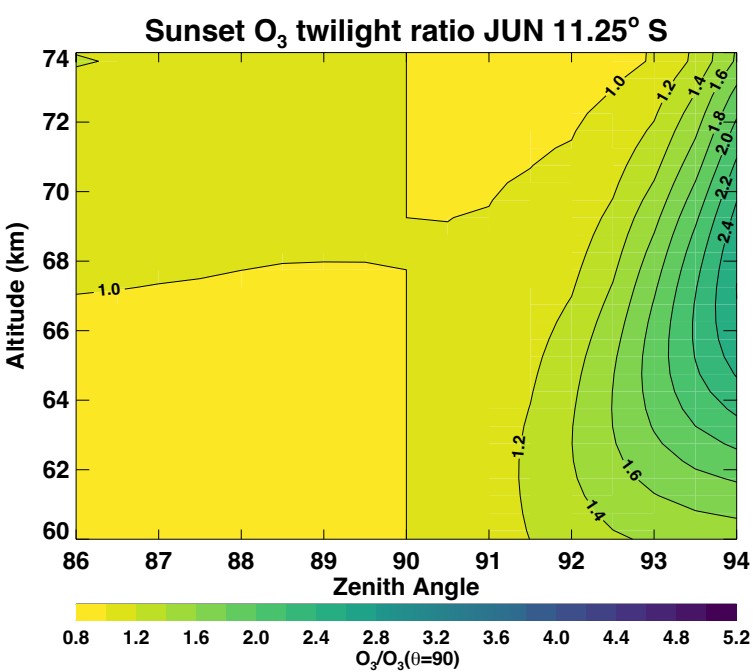


Figure 5.  Ozone twilight ratio, defined as $O_3$ at solar zenith angle $\theta$/$O_3$ at $\theta$=90°, as a function
of SZA and altitude for sunset in June and 11.25° S latitude.

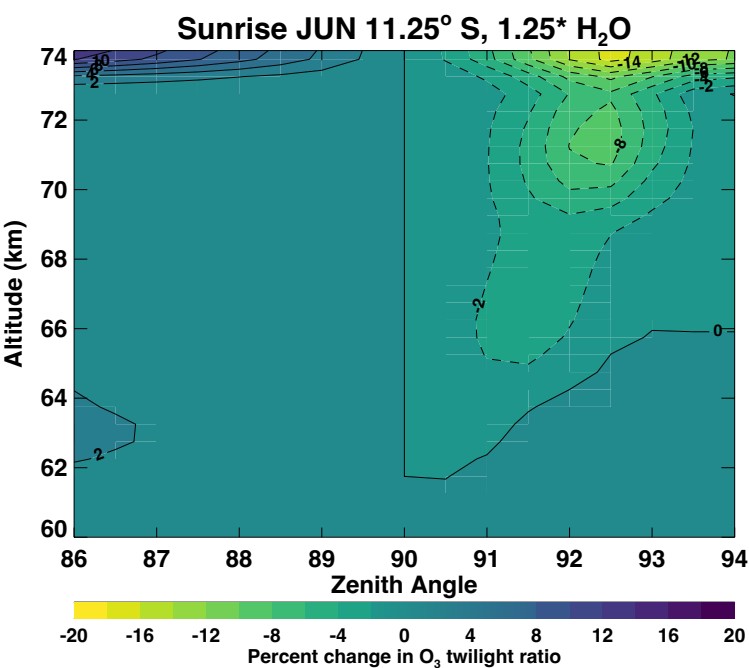


Figure 6.  Percent change in the $O_3$ twilight ratio shown in Fig.4 when the $H_2O$ in the diurnal
model is increased by 25% at all altitudes. This figure corresponds to sunrise at 11.25° S in June.

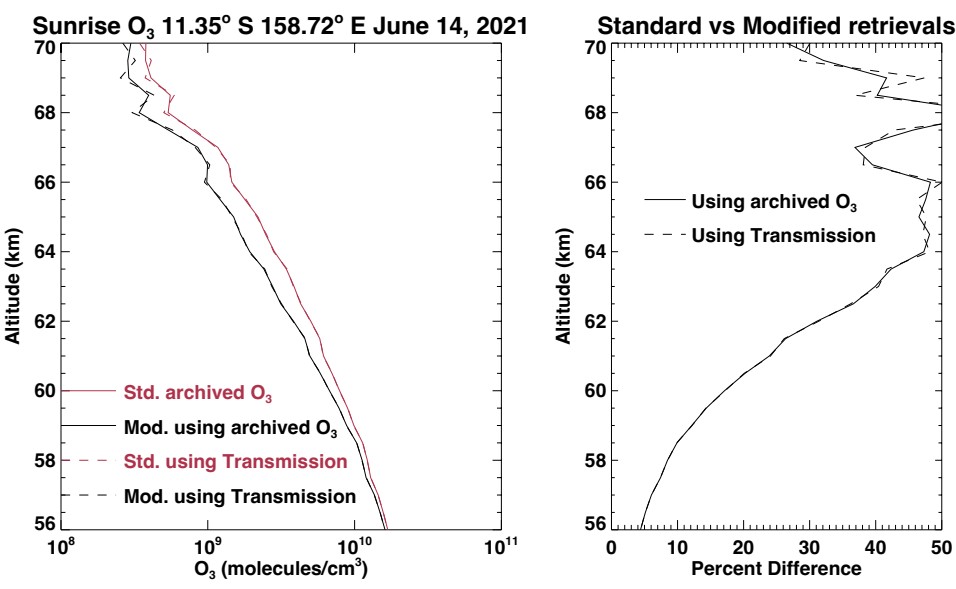


Figure 7.  SAGE III/ISS $O_3$ for a sunrise event at 11.35° S and 158.72° E on June 14, 2021
(Event ID 2021061438SR). (Left panel) Red solid line shows the standard SAGE III retrieval,
and the black solid line represents the retrieval including the diurnal variations along LOS. The
dashed lines represent the retrievals using the transmission data, the red color for the standard
retrieval and the black denoting the retrieval with diurnal corrections. (Right panel) Percent
difference between the standard retrieval and the one with diurnal corrections; solid line using
the archived standard retrieval of $O_3$ concentration, and the dashed line based on the approximate
retrieval using the transmission data.

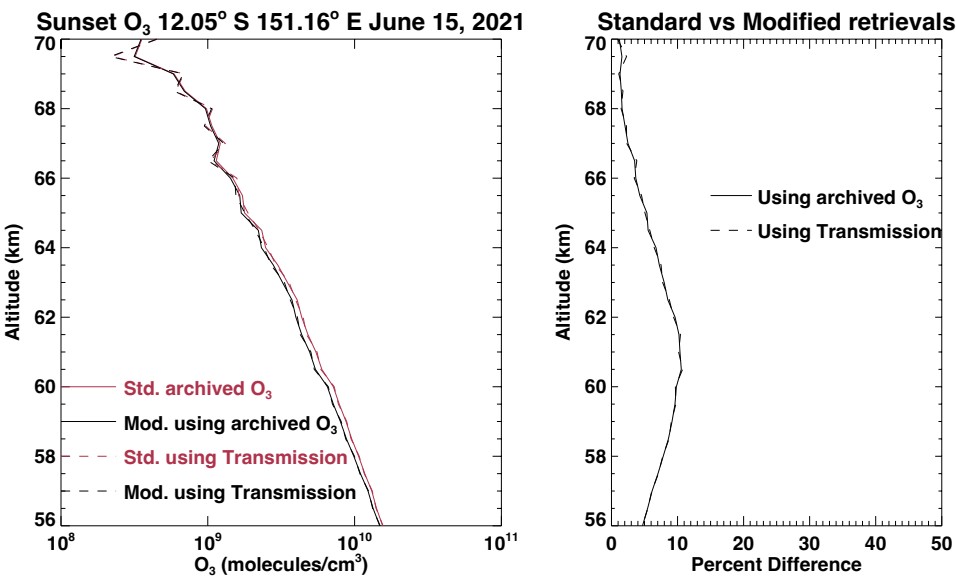


Figure 8. SAGE III/ISS $O_3$ for a sunset event at 12.05° S and 151.16° E on June 15, 2021 (Event

ID 2021061515SS). (Left panel) Red solid line shows the standard SAGE III retrieval, and the

black solid line represents the retrieval including the diurnal variations along LOS. The dashed

lines represent the retrievals using the transmission data, the red color for the standard retrieval

and the black denoting the retrieval with diurnal corrections. (Right panel) Percent difference

between the standard retrieval and the one with diurnal corrections; solid line using the archived

standard retrieval of $O_3$ concentration, and the dashed line based on the approximate retrieval

using the transmission data.


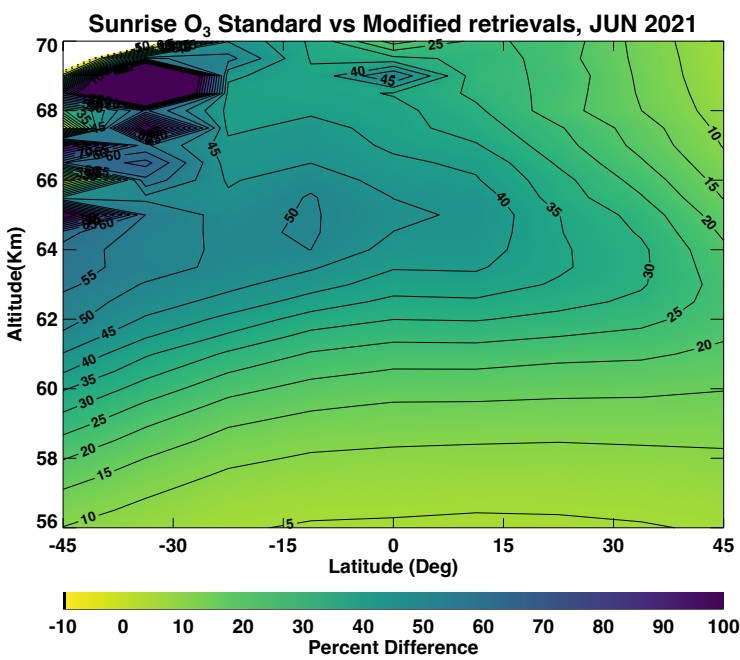


Figure 9. Latitudinal average of the percent difference in sunrise $O_3$ between the standard
(archived) retrieval and a retrieval including diurnal variations along the LOS, as a function of
latitude and altitude for June 2021.


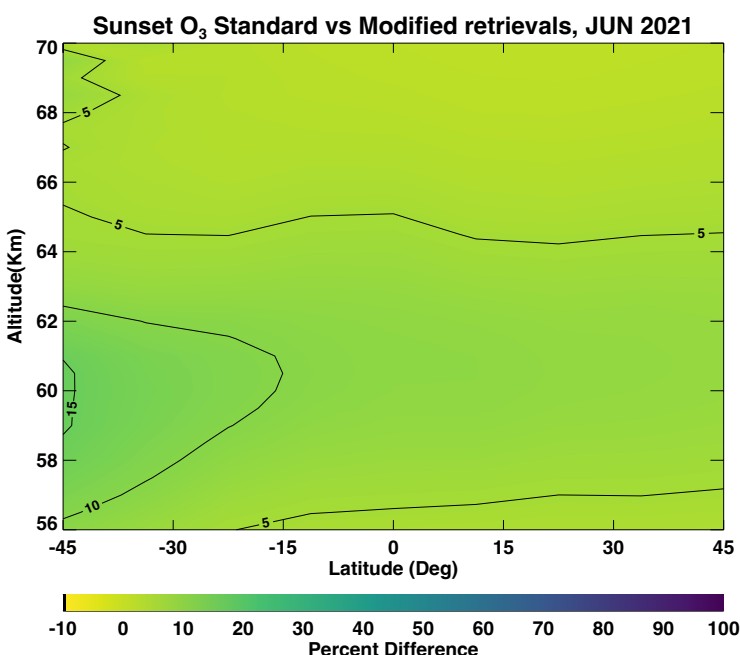


Figure 10.  Latitudinal average of the percent difference in sunset $O_3$ between the standard
(archived) retrieval and a retrieval including diurnal variations along the LOS, as a function of
latitude and altitude for June 2021.

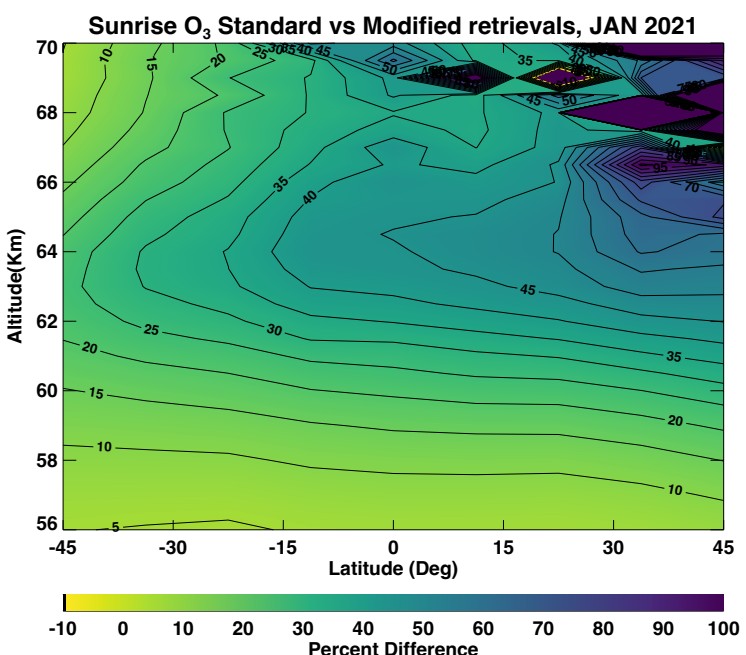


Figure 11.  Latitudinal average of the percent difference in sunrise $O_3$ between the standard
(archived) retrieval and a retrieval including diurnal variations along the LOS, as a function of
latitude and altitude for January 2021.


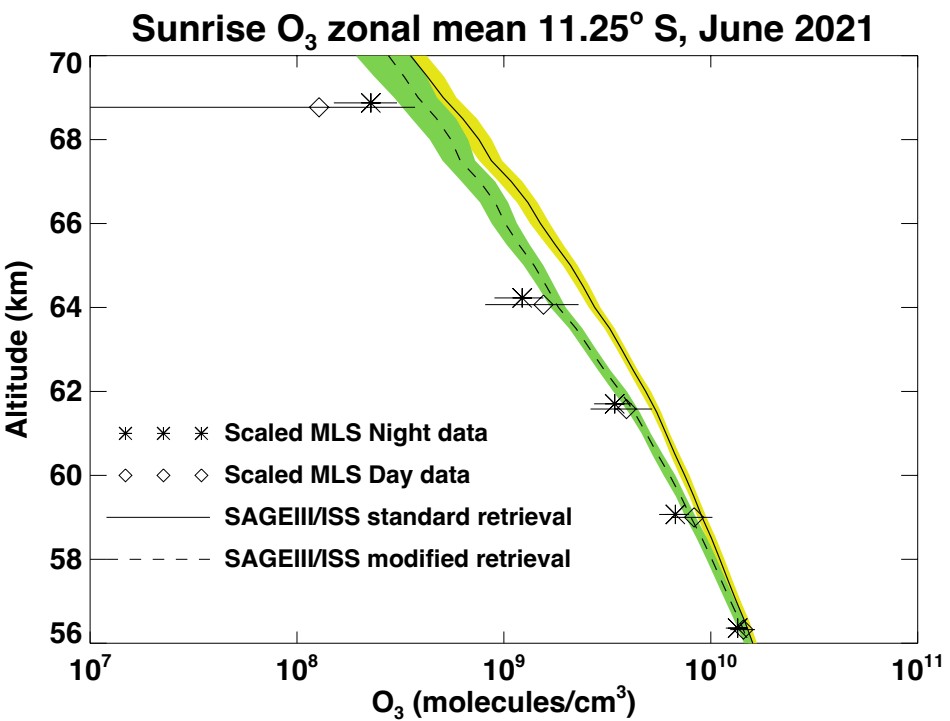


Figure 12.  Comparison of sunrise SAGE III/ISS and MLS mesospheric $O_3$ zonal mean at
11.25° S in June 2021. Solid line – mean of SAGE III/ISS standard retrieval with the standard
deviation shown by the yellow shade; Dashed line – mean of SAGE III/ISS modified retrieval
with the standard deviation shown by the green shade; Asterisks – mean MLS night data scaled
to sunrise; Diamonds – mean MLS day data scaled to sunrise; Horizontal lines represent the
standard deviations.

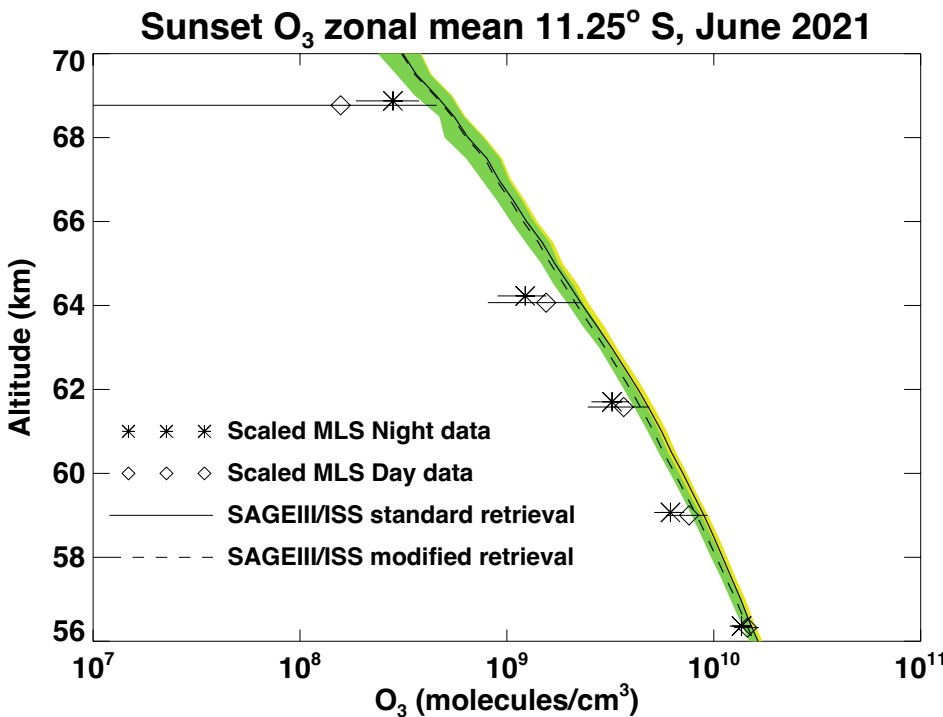

**Sunset O$_3$ zonal mean 11.25$^o$ S, June 2021**


Figure 13.  Comparison of sunset SAGE III/ISS and MLS mesospheric O$_3$ zonal mean at
11.25° S in June 2021. Solid line – mean of SAGE III/ISS standard retrieval with the standard
deviation shown by the yellow shade; Dashed line – mean of SAGE III/ISS modified retrieval
with the standard deviation shown by the green shade; Asterisks – mean MLS night data scaled
to sunset; Diamonds – mean MLS day data scaled to sunset; Horizontal lines represent the
standard deviations.

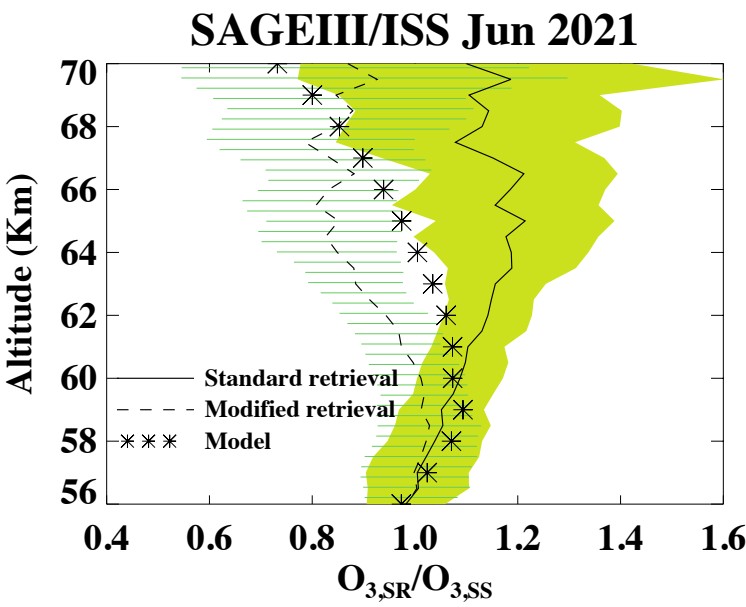


Figure 14. Vertical profile of $O_3$ sunrise to sunset ratio in June 2021. Nearly collocated 10 pairs
of sunrises (mean latitude 10.46° S) and sunsets (mean latitude 10.27° S) data are used for this
plot. Solid line shows the mean ratio from standard (archived) retrieval and the green shade
represents the standard deviation; Dashed line shows the mean ratio from the retrieval including
diurnal variations along the LOS and the horizontal lines represent the standard deviation. The
asterisk symbols are the ratios from diurnal photochemical calculations at 11.25° S for June.