# Peer review of "Solar occultation measurement of mesospheric ozone by 1 SAGE III/ISS: Impact of variations along the line of sight 2 caused by photochemistry 3 4 5 Murali Natarajan1, Robert Damadeo1, David Flittner1 6 1 Science Directorate, NASA Langley Research Center, 21 Langley Blvd., Mail Stop 401-B, 7 Hampton, VA 23681, USA. 8 Correspondence to: Murali Natarajan (murali.natarajan@nasa.gov) 9 Abstract. Twilig"

_Atmospheric Measurement Techniques, 2022_

## Referee Comment (RC2)

**Referee Report to "Solar occultation measurement of mesospheric ozone by SAGE III/ISS: Impact of variations along the line of sight caused by photochemistry" by Murali Natarajan et al.**

The manuscript investigates the influence of the diurnal variation of the mesospheric ozone on the retrievals from SAGE III/ISS measurements and suggests a correction algorithm to account for the diurnal variation in the retrieval of the vertical profiles of the mesospheric ozone. The topic of the manuscript is certainly interesting and has a scientific importance. In general the manuscript is written clearly and concise. A great disappointment for me, however, is the fact that the authors did not make any attempt to compare their results to any independent measurements. A significant difference between the standard retrievals and those corrected for the diurnal variation of ozone is found but the reader is forced to believe that the new results are better and the implementation of the correction scheme is correct. In my opinion, some comparisons with independent measurements need to be added before the manuscript can be published.

**Minor comments**

- Line 39: "longer path length" - longer in comparison to what?

- Line 99: "in this altitude" - please specify the altitude more precise

- Lines 99–102: It is not quite clear how the discussion about the odd oxygen is related to ozone, please clarify.

- Lines 103–105: "At higher altitudes..." - this sentence provides no useful information. Please either delete this sentence or be more specific with respect to altitudes and diurnal behavior.

- Line 129: Please spell out "LaRC"

- Line 133: "positive direction ..." - please provide this information also in figure caption

- Line 134: here and below it is more appropriate to use the term "solar zenith angle" instead of "zenith angle", although I agree it is the same for the occultation geometry.

- Lines 346–350: The discussion about dynamical issues is out of place here. It was not mentioned in the paper and thus should not occur in the summary.

- Figure 1: In the light of the discussion in the manuscript, the figure needs a second x-axis showing the solar zenith angle.

**Technical corrections**

- Lines 25-26: "...the impact of the twilight variations is to increase the optical depth..." - maybe you meant "corrections" rather than "variations".

- Line 138: "is readily seen" $\longrightarrow$ "are readily seen"

---

## Author Comment (AC1)

Response to comments from Referee 1. (Italicized)

In this paper, a correction is discussed for the strong changes of ozone in the (lower) mesosphere along the line-of-sight of a solar occultation measurement due to changing solar zenith angle around the terminator. The method is developed using a 1D chemical box model and applied to as a correction factor to the retrieval of ozone from SAGE III/ISS. Corrected and uncorrected profiles ozone profiles are compared, and differences of up to 50% are found for sunrise observations above 64 km, around 10% for sunset observations around 62 km. Similar corrections are used for the retrieval of stratospheric NO and NO2 but are apparently not widely used for mesospheric ozone. Considering the large differences particularly for the sunrise data, it is certainly good to address this issue. The paper is generally very well written, but I have a few points which should be addressed listed below.

*We thank the reviewer for the comments and suggestions. Our responses to the comments are shown below in italics. We hope the revised manuscript addresses the comments to the satisfaction of the reviewer.*

Line 22/23, also lines 46-48, line 200: (1) while you allow for variation of ozone with the solar zenith angle along the line of sight, you still have to make the assumption that Ox = O + O3 is constant along the LOS, is that correct? Can you state this a bit more clearly? (2) Were the model experiments carried out for different latitudes / seasons, or just for the one example shown (tropics)?

(1) *In this analysis we don't assume that $O_x$ is constant along the LOS. We do use the fact that the integrated column of $O_3$ along the LOS is the same for the standard and modified retrievals (Equation 2 in the text). The mesospheric $O_3$ column along the LOS in the solar occultation experiment comprises $O_3$ concentrations corresponding to different SZA from every level above the tangent altitude. The $O_x$ in each of these levels is different and the partitioning into O and $O_3$ is also different based on the local SZA. The one-dimensional time dependent model doesn't assume constant $O_x$. Both O and $O_3$ are independent variables in the model. In the lower mesosphere, the chemical lifetime of $O_x$ is shorter than a day. The production of $O_x$ from the photolysis of $O_2$ during the day is balanced by the loss of $O_x$ from $O_x$ and $HO_x$ reactions integrated over the diurnal cycle. We have added an appendix showing the photochemical scheme used in the model.*
(2) *The diurnal model calculations were done for each month at 11 latitude bands 11.25° wide from 56.25°N to 56.25 °S. The diurnal factors were interpolated over latitudes and calendar days.*

Line 30: see my comment below (line 311-312) about the sunrise to sunset ratio as shown in Figure 11

*Please see the response below (line 311-312)*

Line 88-89: considering the model results are really an essential part of the paper, a more concise description would be appropriate. At the very least you should mention which species are considered, and how the model is initialized, which certainly has some impact on the model results (i.e., how much H2O or HOx will have a big impact on daytime ozone). A list of the photochemical reactions considered would be good as well, maybe in an Appendix.

*We appreciate this suggestion from the reviewer. We have expanded the description of the model and added the mesospheric photochemical reaction scheme used in the model as an appendix. We recognize that the absolute $O_3$ concentrations are dependent on factors such as the abundance of $H_2O$ used in the model. We point out that the variation in $O_3$ near the terminator normalized to its value at SZA of $90°$ is very robust and the twilight ratio can be used to make the correction required in the retrieval. We have also added a figure illustrating the sensitivity of the $O_3$ twilight ratios to a 25% increase in $H_2O$.*

Line 92: please state the altitude range here, and explain why it is restricted to 58-74 km. This information is provided further down, but really belongs here.

*We state the altitude range in this section of the revised version.*

Lines 92-93 and lines 102-103: these lines appear to be in contradiction. The figure shows constant ozone (presumably constant Ox) during night, in agreement with lines 92-93. Or do you mean "around sunset" in line 103?

*Around sunset, there is a net loss of Ox which continues until all the O is transformed to $O_3$. Within a few hours after sunset $O_x$ and $O_3$ reach a steady value. We have revised the line 103 to make this clear.*

Line 103-104: as results for higher altitudes are shown later, you should also show model results from these altitudes.

*We have decided to remove the results for higher altitudes because of the noisy data. The large twilight ratios above 73 km combined with noisy data add to the uncertainty of the results for higher altitudes. We have also revised the text in line 104 (New line 122)*

Line 105-106: however, you do show results from 70-100 km for the O3 day/night ratio, so you maybe should show results and discuss this region here as well.

*We now limit our attention to the region below 70 km and do not show the results from the upper regions.*

Line 268, discussion of the impact of the twilight correction as shown in Figure 8: you stated before that data above 70 km are very noisy, and this is presumably the reason for the very patchy structure with occasionally high values (100%). First of all – are you certain there are no NaNs or negative / unrealistically low values in this sample? Considering the high noise, it would make sense not to show the data above 70 km as you did for other properties. However, if

you want to show them, you should average over larger samples, either by increasing the latitude bins above 70 km, or by calculating a running average above 70 km.

*We have decided not to show the data and the results above 70 km because of the noisy structure. O₃ profile reaches a minimum in the 75 to 80 km region and the uncertainty in the data is high. We have retained the negative values, but we removed the data points with large filled-in values representing low confidence in the retrieval.*

Line 305-308: can you provide error bars, i.e., the standard error of the mean, for the corrected and uncorrected values?

*We have revised the plot to show the mean and the standard deviation for both the standard and modified retrievals.*

Line 311-312: considering the large quantitative differences between the theoretical values and the corrected and uncorrected values you could argue with as much justification that the uncorrected values are in better agreement with the theoretical values as they seem to agree better quantitatively in the lower altitudes. A clear statement which fits better seems difficult here. However, it might be possible to provide a more robust statement if error bars were provided.

*We emphasize that this figure only shows a qualitative improvement achieved by the modification to the retrieval scheme. The sunrise to sunset ratio should exhibit a decreasing value with altitude in this region based on the known photochemistry. The modified retrieval yields such a profile though the standard deviations are large enough to overlap.*

Line 345-346: see my comment above to lines 311-312.

*See response above.*

**Citation**: https://doi.org/10.5194/amt-2022-266-RC1

---

## Author Comment (AC2)

Response to the comments from referee 2 (Italicized)

Referee Report to "Solar occultation measurement of mesospheric ozone by
SAGE III/ISS: Impact of variations along the line of sight caused by photo-
chemistry" by Murali Natarajan et al.
The manuscript investigates the influence of the diurnal variation of the mesospheric ozone
on the retrievals from SAGE III/ISS measurements and suggests a correction algorithm to
account for the diurnal variation in the retrieval of the vertical profiles of the mesospheric
ozone. The topic of the manuscript is certainly interesting and has a scientific importance.
In general the manuscript is written clearly and concise. A great disappointment for me,
however, is the fact that the authors did not make any attempt to compare their results to
any independent measurements. A significant difference between the standard retrievals
and those corrected for the diurnal variation of ozone is found but the reader is forced to
believe that the new results are better and the implementation of the correction scheme
is correct. In my opinion, some comparisons with independent measurements need to be
added before the manuscript can be published.

*We thank the reviewer for the comments. We have revised the manuscript to consider these
comments and we hope this version will satisfy the reviewer.*

*We have included a section discussing intercomparison with other data. Mesospheric O$_3$
measurements under similar conditions regarding the time of the day are available only from
other solar occultation experiments like HALOE and ACE-FTS. Irrespective of whether these
experiments include the corrections for the twilight variations of O$_3$ in their retrieval schemes or
not, they cannot independently validate the need for or the magnitude of such a correction.
Other experiments such as MLS and ground-based microwave radiometry provide mesospheric
O$_3$ data. We have shown a comparison MLS data with SAGE III/ISS for a selected latitude band
and period in June 2021. We have chosen to compare mean profiles taken over a 3-day period.
Diurnal variation in O$_3$ makes it necessary to convert the MLS day or night measurements to
SZA of 90° during sunrise or sunset. For this, we used the diurnal photochemical model results.
We have shown the results from this comparison and discussed the implications. We feel that the
use of the diurnal model for both the modified SAGE III/ISS O$_3$ retrieval and the conversion of
MLS O$_3$ makes this comparison strictly not an independent validation of the need for twilight
correction of the solar occultation retrieval. In the absence of any other independent data taken
at SZA of 90°, we realize that this is probably the best we can do. The main emphasis of this
paper is not the validation of SAGE III/ISS mesospheric O$_3$ measurements but to highlight the
cause for a bias in the retrieval and suggest a correction scheme.*

Minor comments
• Line 39: "longer path length" - longer in comparison to what?

*Reference to the path length is removed.*

• Line 99: "in this altitude" - please specify the altitude more precise

*Revised the sentence to indicate the altitude range where $O_x$ has a diurnal variation.*

• Lines 99–102: It is not quite clear how the discussion about the odd oxygen is related to ozone, please clarify.

*The odd oxygen variation and its relation to $O_3$ diurnal changes have been added.*

• Lines 103–105: "At higher altitudes..." - this sentence provides no useful information. Please either delete this sentence or be more specific with respect to altitudes and diurnal behavior.

*This sentence has been removed since we have decided to focus our discussion on the region below 70 km.*

• Line 129: Please spell out "LaRC"

*Done*

• Line 133: "positive direction ..." - please provide this information also in figure caption

*Revised the figure caption to include more information about the axis.*

• Line 134: here and below it is more appropriate to use the term "solar zenith angle" instead of "zenith angle", although I agree it is the same for the occultation geometry.

*Done*

• Lines 346–350: The discussion about dynamical issues is out of place here. It was not mentioned in the paper and thus should not occur in the summary.

*We have deleted the sentence about dynamical issues from the Summary.*

• Figure 1: In the light of the discussion in the manuscript, the figure needs a second x-axis showing the solar zenith angle.

*A second X-axis showing the solar zenith angle is added to the Figure 1.*

Technical corrections
1
• Lines 25-26: "...the impact of the twilight variations is to increase the optical depth..." - maybe you meant "corrections" rather than "variations".

*The sentence has been modified to make it clearer.*

• Line 138: "is readily seen" −→ "are readily seen

*Corrected.*

---

## Author Response (AR2)

**Response to the editor's comments. (*italicized*)**

**Comments to the author:**
Dear Authors, I had a look at the tracked version, and here are some comments.

1. line 125 :Give some information about chemical and dynamical time step intervals in seconds

*I have added two sentences about the algorithm used for the model and the time steps (line 111). The diurnal model is a one-dimensional photochemical model.  There is no separate dynamical time step.  Species with a variety of chemical time constants make the system very stiff and we use an automatic solver designed for such systems. The maximum timestep is set at 600 seconds and the algorithm adjusts the time step to very low values (< 1 millisecond near the day/night terminator). We have added the reference to the numerical solver used by the diurnal model.*

**Figure captions should be independent and should provide detail description about the plots.**

*Figure captions have been revised, where required, to provide full description of the plots.*

Some places in the manuscript sound like a cation, please edit those lines
e.g.
2. line 141: Delete - "The upper X-axis shows the corresponding SZA. It is correct to include it caption

*Deleted the sentence, as suggested.*

3. line 194 : Delete "The X-axis shows ..."-- Caption should include all the relevent information

*Revised the sentence.*

4. line 342 : The right panel

*Deleted the redundant sentence.*

5. Line 538: Do you mean O3 at 64 km at lower latitudes?

*Yes. The text has been corrected.*

Figure 7 & 8 Caption: include latitude/longitude information in the caption.

*Latitude / Longitude information is provided in both the figure captions.*